# The Burden of Hospitalization and Rehospitalization Among Patients Hospitalized with Severe Community-Acquired Bacterial Pneumonia in the United States, 2018–2022

**DOI:** 10.3390/antibiotics14070642

**Published:** 2025-06-25

**Authors:** Marya D. Zilberberg, Mike Greenberg, Valentin Curt, Andrew F. Shorr

**Affiliations:** 1EviMed Research Group, LLC, Goshen, MA 01032, USA; 2Eagle Pharmaceuticals, Inc., Woodcliff Lake, NJ 07677, USA; mgreenberg@eagleus.com (M.G.); vcurt@eagleus.com (V.C.); 3Washington Hospital Center, Washington, DC 20010, USA; andrew.shorr@gmail.com

**Keywords:** bacterial pneumonia, burden of illness, community-acquired pneumonia, hospitalization, severe CAP

## Abstract

**Background**: Community-acquired bacterial pneumonia (CABP) is a common and costly cause of hospitalization. Although severe CABP (sCABP) occurs in 10–25% of all pneumonia hospitalizations, little generalizable data examine its characteristics and outcomes or hospital resource utilization. **Methods**: We conducted a retrospective single-group cohort study of adults within the IQVIA hospital Charge Data Master, 2018–2022. We identified CABP via an ICD-10 code algorithm and sCABP was defined as an episode requiring ICU care. We examined baseline characteristics and outcomes, including mortality, costs, and readmission rates. We developed models to identify risk factors associated with readmissions. **Results**: Among 24,149 patients with sCABP, 14,266 (58.4%) were ≥65 years old and 55.2% were male. The majority were hospitalized in large (300+ beds, 50.9%), urban (91.9%) teaching (62.7%) institutions in the US Southern region (52.3%). The mean (SD) Charlson Comorbidity Index was 1.35 (2.33). The most common comorbidities were hypertension (16.7%), diabetes mellitus (15.7%), and chronic obstructive pulmonary disease (COPD) (12.9%). Hospital mortality was 15.9%. The mean (SD) hospital length of stay (LOS) and costs were 13.6 (12.1) and USD 91,965 (USD 133,734), respectively. An amount of 20% required a readmission within 30 days. Readmission was most strongly associated with older age and the presence of select comorbidities (diabetes mellitus, congestive heart failure, and COPD), each with an odds ratio > 1.4 and 95% confidence intervals excluding 1.0. **Conclusions**: Patients with sCABP comprise a large population with high mortality and 30-day readmissions. The intrinsic factors related to the latter lend themselves to early recognition and aggressive efforts at reducing complications.

## 1. Introduction

Community-acquired pneumonia (CAP) remains one of the most common and costly causes of hospitalization in the United States and worldwide. In the United States, there are over 1.4 million admissions annually for pneumonia, with 10% short-term mortality. Financially, CAP costs the healthcare system over USD 14 billion each year [1,2,3,4,5]. Beyond pneumonia itself, CAP can result in a slew of secondary hospital-acquired complications (HACs) such as *Clostridioides difficile* infection (CDI), a reportable outcome under the US Centers for Medicare and Medicaid Services (CMS) HAC Reduction Program, which occurs in up to 3% of patients with CAP [6,7,8].

In addition to the initial hospitalization, CAP represents the fifth most common cause of readmission in the United States [9]. Multiple studies have documented that 12–20% of patients discharged alive from their index pneumonia episode require rehospitalization within 30 days following discharge [10,11]. Because of this, pneumonia falls under the CMS Hospital Readmissions Reduction Program, which places financial penalties on hospitals with above-average rates of this readmission [12,13,14,15].

While mortality and morbidity vary according to various underlying host and pathogen factors, these remain highest among those presenting in dire conditions and who are severely ill. For this reason, more attention has recently been paid to severe CAP (sCAP), whose attendant mortality is closer to 20%, much higher than in CAP overall [16,17]. Multiple process-of-care variables can help limit the burden of sCAP, such as the timely administration of appropriate antimicrobial treatment. Other factors, perhaps less susceptible to modification, may nonetheless aid in the early identification of patients at highest risk of poor outcomes.

Although sCAP represents 10–25% of all pneumonia admitted to the hospital, few large, generalizable, recent studies have addressed this population. As a result, little is known about the characteristics and outcomes of patients with sCAP or their hospital resource utilization (HRU) in particular [16,17]. We conducted a descriptive study to address this gap and to develop a predictive model for hospital readmissions.

## 2. Results

Among 24,422 patients who met the inclusion criteria, 14,266 (58.4%) were ≥65 years old and 55.2% were male (Table 1). The proportion of annual hospitalizations was lowest in 2019 (prior to the COVID-19 pandemic) and highest in 2020, the year of the pandemic. There was seasonality to the sCABP admissions, with the highest percentage admitted in January (12%) and the lowest in October (6%) (Appendix A, Appendix A). The majority of hospitalizations took place in the Southern census region (52.3%), in large (300+ beds, 50.9%) urban (91.9%) teaching (62.7%) institutions, and the vast majority were admitted through the Emergency Department (85.7%). The comorbidity burden in the cohort was moderate, with the mean (standard deviation, SD) Charlson Comorbidity Index equaling 1.35 (2.33) (Table 1). The most common comorbidity was hypertension (16.7%), followed by diabetes mellitus (15.7%); chronic lung disease was also frequent (12.9%) (Appendix A, Appendix A). The most common medication received prior to index admission was a cephalosporin antibiotic (12.3%), followed by a respiratory inhaler (10.9%) (Appendix A, Appendix A).

Treatment characteristics are listed in Table 2. The most common monotherapy for sCABP was a cephalosporin (36.2%), while a cephalosporin and a macrolide was the most frequent combination treatment (22.9%). A glycopeptide was used in 43.1% of patients, and a penicillin with a beta-lactamase inhibitor was used in 32.4% of patients. Fluoroquinolones as monotherapy were administered in only 12.4% of the cohort.

The unadjusted outcomes are listed in Table 3. Hospital mortality was 15.9%, and CDI developed among 1.2% of all patients. The mean (SD) ICU and hospital LOS were 8.7 (9.6) and 13.6 (12.1) days, respectively. The corresponding costs were USD 23,847 (USD 36,760) for the ICU and USD 91,965 (USD 133,734) for the hospital stays. Readmissions within 30 and 90 days after discharge from the index hospitalization among survivors were 19.9% and 29.9%, respectively, of whom a small minority were admitted for the treatment of CABP (Table 3) (see Appendix A, Appendix A for most common readmission diagnoses). The majority (86.9% and 71.3% at 30 and 90 days, respectively) of all readmitted patients had only a single readmission (Figure 1). The LOS for the first readmission and the mean LOS across all readmissions within the 30-day and 90-day time frames were numerically higher than the LOS for the index admission. The corresponding mean (SD) costs for these readmissions trended lower than those for the index hospitalization (USD 61,072 [USD 102,604] for 30- and USD 72,117 [USD 113,421] for 90-day total readmissions) (Table 3).

Readmissions within 30 and 90 days after discharge from the index hospitalization among survivors of sCABP were 19.9% and 29.9%, respectively, of whom a small minority were admitted for the treatment of CABP (Table 3; see Appendix A, Appendix A for most common readmission diagnoses). The majority (86.9% and 71.3% at 30 days and 90 days, respectively) of all readmitted patients had only a single readmission.

In a logistic regression to derive risk factors for 30- and 90-day readmissions, several risk factors were identified (Table 4). Advancing age, history of diabetes mellitus, congestive heart failure, and COPD were most strongly associated with these outcomes. Moreover, each additional day in the ICU during the index hospitalization increased the odds of readmission by approximately 1%.

## 3. Discussion

We demonstrate that a large number of patients admitted for CAP meet criteria for sCABP. Strikingly, the hospital mortality for these patients is 16%. The costs of these hospitalizations are high, nearing, on average, USD 100,000/case. In addition, survivors of a hospitalization for sCABP are at high risk for readmission, both at 30 days (20%) and 90 days (30%). While these readmissions are rarely for the further treatment of pneumonia per se, they add considerable costs to the overall care of these patients.

A large cohort study in Louisville, Kentucky, enrolled nearly 7500 patients with CAP, of whom 26% required treatment in the ICU [16]. The median age in the ICU group was identical to that in our cohort (67 years), as was hospital mortality (17%). While the Kentucky cohort did not examine costs or readmission outcomes, their study design allowed the investigators to project their results to national estimates. Their estimate for the annual volume of CAP requiring ICU care added up to over 350,000 patients. Despite the fact that a small minority (14%) of sCAP is bacterial in nature, clinicians continue to treat most patients presenting to the hospital with pneumonia symptoms as though they had CABP, and this is particularly true for those with severe disease [5,17,18]. For this reason, our estimates of health resource utilization, despite being limited to those with sCABP, likely apply to most sCAP, irrespective of whether a pathogen has been identified. Thus, applying our 20% 30-day readmission rate to estimates from Cavallazzi et al. yields 75,000 potential readmissions in the United States annually [16]. An estimated cost of USD 61,000 per event suggests that annual readmissions may result in costs totaling up to USD 4.5 billion per year, with this being in addition to the USD 32 billion aggregate financial costs related to the index admissions.

Another recent retrospective cohort study by Haessler et al., relying on the Premier Healthcare Database, examined patients with sCAP as defined by two major criteria: the need for mechanical ventilation or treatment with vasopressors [17]. These investigators found that nearly 90% of the patients identified, not surprisingly, required treatment in the ICU. While the overall hospital mortality in this group was 22%, the mortality at 14 days was 18%. The attendant median hospital costs were USD 20,000. Our mortality findings comport with those reported by Haessler et al., particularly given that our median hospital LOS was 12 days. The median hospital costs in our study, however, were over double those noted by Haessler et al. [17]. One reason for this may be the fact that the IQVIA database leans more toward commercially insured persons and less toward the Medicare population. It is well established that commercial insurance reimbursements to hospitals are on average double those made by Medicare [19].

An additional finding of our study is that the incidence rate of hospital-onset CDI was 1.2%. Prior investigations have noted a substantial risk of CDI development among patients hospitalized with CAP. This is not surprising, given that pneumonia is one of the most common reasons to receive antibiotics, and the classes of drugs indicated for its treatment are associated with an increased risk of CDI development. In a cohort of nearly 150,000 patients with CAP in the United States, Patel et al. reported a CDI rate of 0.5% [7]. Although the estimate of incident CDI from Patel et al. was less than half of the estimate from this study, their population consisted of all CAP severities. Indeed, among CAP patients, a number of risk factors were discovered to be associated with a higher risk of CDI, including the need for an ICU stay (odds ratio of 1.43). Additionally, our CDI definitions differed in that Patel et al. relied on a positive laboratory test, whereas we defined a case of CDI by an ICD-10 code that was not present on admission. The former approach may be more restrictive than the latter, yielding a lower rate of CDI identification. At the same time, hospital-acquired complications may be under-coded, making the reliance on ICD-10 codes a less sensitive way to detect CDI. Chalmers et al. in the United Kingdom, in contrast, reported an incident CDI rate of 3.2% among patients hospitalized with CAP [6]. In this subanalysis of a prospective cohort study of 1883 patients admitted with CAP, CDI diagnosis was based on a positive laboratory test. While ICU admission occurred in under 10%, a full third of the cohort had severe pneumonia based on the Pneumonia Severity Index. Notably, the mortality among CAP patients who contracted CDI (21.3%) rose by nearly 3-fold relative to those without it (8.6%). Importantly, this study identified several modifiable risk factors associated with CDI, including the duration of antimicrobial therapy, as well as overall hospital LOS.

A systematic review focusing on resource utilization by patients whose illness is complicated by CDI development estimated the 6-month CDI attributable costs to equal USD 24,000, underscoring the economic rationale to reduce patients’ exposure to this infection in addition to the clinical one [20]. The incidence of CDI that we observed in the current analysis was 1.2%, falling within the reported range and a number that may underrepresent its actual occurrence. Since patients with sCABP may be even more prone to prolonged antibiotic regimens and a longer LOS than the overall CAP population, future research, using data where both testing and treatments are available to improve the specificity of detection, should focus on defining risk factors in sCABP with the aim of reducing the burden of CDI in these patients.

Our study’s biggest strength is that it is a large multicenter study, and thus highly generalizable to both patients and various institutions. At the same time, there are a number of important limitations. Because the database consists of administrative coding, our cohort definition, as well as some of the hospital events and outcomes, are prone to misclassification. Using the locus of care in the ICU as the defining factor for sCABP introduces a degree of heterogeneity in our identification of the cohort, given the inter-hospital variability in ICU admission practices. That is, some patients with more severe disease may be cared for outside of an ICU at select hospitals with a higher ICU bed demand, while at other hospitals, more moderately ill patients may be admitted to an ICU. It would have been more helpful to have employed specific, patient-focused criteria for severe pneumonia, such as the need for mechanical ventilation or vasopressors [21]. Unfortunately, the current database lacks this information. Given that the minor criteria for sCABP defined by Infectious Diseases Society of America/American Thoracic Society consensus guidelines involve physiologic and laboratory factors, which are also missing from IQVIA data, the need for ICU admission represented the only way to define sCABP in this study. At the same time, the high rates of treatment with glycopeptides and extended-spectrum penicillins suggest that clinicians believed these cases truly represented a clinical presentation consistent with the criteria that define sCABP [17]. Importantly, the outcomes we examined, such as hospital mortality and 30- and 90-day readmissions, are less prone to being misclassified, with one important caveat: it is possible that we may be missing data on survivors of the index hospitalization who were readmitted to a hospital outside the IQVIA hospital universe. Thus, we have likely underestimated the magnitude of the problem of 30- and 90-day readmissions. In this sense, our estimate of the readmission rate represents a lower bound for this event and indicates that the need for readmission in sCABP is almost certainly greater. Bias is another important threat to the validity of observational data. To minimize that, we developed a priori definitions and analysis plans and enrolled all consecutive patients meeting predefined inclusion criteria. Given that this is a single-group cohort analysis, confounding was not an issue. However, our models may contain parameters that merely serve as surrogates for other factors that are unavailable in the current data, with one example being the more clinical features of sCABP, such as the need for mechanical ventilation or vasopressors.

In summary, we have demonstrated that there is a large population of hospitalized patients with sCABP who have poor clinical outcomes and high hospital mortality. Their costs are substantially higher than those reported for non-severe CABP, as is the burden of their rehospitalizations, both at 30 and 90 days after initial discharge. The readmissions are related mostly to intrinsic factors, such as advanced age and certain comorbidities, that lend themselves to early recognition and focused efforts to prevent HACs, such as CDI. Given the mortality and morbidity of sCABP, there is a surprising dearth of research into this population. Future studies should focus on the validity of definitions and outcomes for sCABP, as well as evaluate strategies to prevent its attendant mortality, morbidity, and economic consequences.

## 4. Methods

### 4.1. Ethics Statement

Because this study used fully de-identified administrative data, it was exempt from ethics review under US 45 CFR 46.101(b)4 [22].

### 4.2. Study Design and Patient Population

We conducted a multicenter, retrospective, single-group cohort study of adult (≥18 years) hospitalized patients with bacterial sCAP (sCABP). CABP inpatients were identified by the admitting or principal discharge diagnosis of CABP, or a secondary discharge diagnosis of CABP in the face of a primary diagnosis of sepsis or acute respiratory failure. The case identification approach relied on International Classification of Diseases-10 (ICD-10) codes, excluding a diagnosis of non-bacterial pneumonia during index hospitalization (see Appendix A, Appendix A). Severe CABP was defined by the need for an intensive care unit (ICU) stay during the index hospitalization. To ensure that the index CABP hospitalization was not a repeat episode, we excluded all patients with any pneumonia within 90 days prior to index admission. Additional exclusion criteria were transfer from another hospital, missing or invalid year of birth, missing gender, or missing total charge or no revenue charges.

### 4.3. Data Source and Time Frame

The data for this study were derived from IQVIA’s hospital Charge Data Master (CDM) [23]. It comprises data sourced from 450 hospitals representing 8 million inpatient visits and 105 million outpatient visits per year. The database contains standard information from hospital claims, such as patient age, gender, principal and secondary diagnoses, and procedures. Race/ethnicity data are not available in this database.

The time frame for the analyses spanned from 1 October 2017 through 31 January 2022 (Appendix A, Appendix A). To be included, a patient had to have a hospitalization for CABP with admission dates between 1 January 2018 and 30 September 2021 (selection window), a discharge date no later than 31 October 2021, evidence of ICU admission during the hospitalization, and data available during the 90 days after the index hospitalization. This window was used to allow for a 90-day wash-in period without a pneumonia episode, and a 90-day follow-up period to examine the incidence of 30- and 90-day readmissions.

### 4.4. Baseline Measures

Baseline factors included both hospital (geographic area, size, urbanicity, academic affiliation) and patient (demographics, clinical) characteristics. We computed a Charlson Comorbidity Index score as a measure of the burden of chronic illness [24]. In addition to standard factors, we evaluated for certain medications administered within 90 days prior to index hospitalization (immunosuppressants/chemotherapy, inhalers for lung diseases, antibiotics), antimicrobial treatment for sCABP, and underlying conditions, such as immunodeficiency and other chronic conditions.

### 4.5. Outcome Variables

Mortality served as the primary outcome. Secondary outcomes included hospital and ICU length of stay (LOS) and costs, 30- and 90-day readmissions among survivors of the index admission, and incidence of CDI as a complication. The CDM contains charges which we converted to costs using the hospital-specific cost-to-charge ratios provided by the Agency for Healthcare Research and Quality (AHRQ) through their Healthcare Cost and Utilization Project [25].

### 4.6. Statistical Analyses

We derived summary descriptive statistics for the baseline, treatment, and outcomes variables. In addition, we developed regression models to examine the risk factors for a 30- and 90-day readmission based on baseline, infection, and hospital treatment characteristics. We utilized a stepwise model-building approach, with *p* < 0.10 set as a threshold for inclusion and retention. Collinearity among the variables of interest was evaluated prior to model development, and highly correlated variables were excluded. Statistical significance was set at *p* < 0.05. All analyses were performed in SAS v9.4, Cary, NC, USA.

## Figures and Tables

**Figure 1 antibiotics-14-00642-f001:**
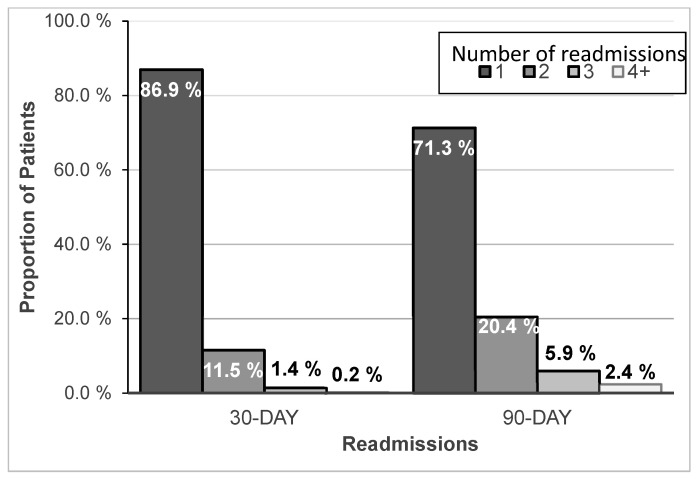
The proportion of SCABP patients with single and multiple readmissions within 30 and 90 days of index discharge.

**Table 1 antibiotics-14-00642-t001:** Baseline characteristics.

	Total N = 24,422	
Characteristic	n	%
Age (years)		
Mean	65.4	NA
SD	14.6	NA
Median	67	NA
Age Group		
2–17 years	0	0.00%
18–34 years	1104	4.52%
35–44 years	1301	5.33%
45–54 years	2381	9.75%
55–64 years	5370	21.99%
≥65 years	14,266	58.41%
Gender		
Male	13,486	55.22%
Female	10,936	44.78%
Geographic region		
Northeast	2791	11.43%
Midwest	1838	7.53%
South	12,762	52.26%
West	7031	28.79%
Payer		
Cash	74	0.30%
Medicaid	1084	4.44%
Medicare Risk	8600	35.21%
Third Party	10,399	42.58%
Other/Unknown	4265	17.46%
Year of the index hospitalization		
2018	5743	23.52%
2019	6531	26.74%
2020	7502	30.72%
2021	4646	19.02%
Number of hospital beds		
1–99 beds	2137	8.75%
100–199 beds	5218	21.37%
200–299 beds	4665	19.10%
300–499 beds	7140	29.24%
500+ beds	5255	21.52%
Unknown	7	0.03%
Hospital urbanicity		
Urban	22,437	91.87%
Rural	1985	8.13%
Hospital teaching status		
Teaching	6373	26.10%
Non-Teaching	15,451	63.27%
Unknown	2598	10.64%
Admitting source		
Emergency department	20,934	85.72%
Referral and transfer	246	1.01%
Routine admission	2590	10.61%
Other	286	1.17%
Unspecified	366	1.50%
Charlson Comorbidity Index (CCI)		
0	15,596	63.86%
1	1816	7.44%
2	1797	7.36%
3+	5213	21.35%
Mean	1.35	NA
SD	2.33	NA
Median	0	NA

SD, standard deviation. NA, not applicable.

**Table 2 antibiotics-14-00642-t002:** Pneumonia treatment characteristics.

	Total N = 24,422	
	n	%
Index treatment antimicrobial class (not mutually exclusive)		
Aminoglycosides	272	1.11%
Antifungals	428	1.75%
Carbapenems	1436	5.88%
Cephalosporins	13,124	53.74%
Fluoroquinolones	3695	15.13%
Folate pathway inhibitors	3	0.01%
Glycopeptide	10,520	43.08%
Macrolides	7925	32.45%
Monobactams	450	1.84%
Monoclonal antibodies	0	0.00%
Other antibiotics	1630	6.67%
Oxazolidinone	354	1.45%
Penicillin	161	0.66%
Penicillin with beta-lactamase inhibitors	7919	32.43%
Polymyxins	11	0.05%
Tetracyclines	1014	4.15%
Treatment type (mutually exclusive)		
Monotherapy		
Aminoglycosides	162	0.66%
Antifungals	736	3.01%
Carbapenems	2448	10.02%
Cephalosporins	8841	36.20%
Fluoroquinolones	3039	12.44%
Folate pathway inhibitors	1	0.00%
Glycopeptides	2512	10.29%
Macrolides	1433	5.87%
Monobactams	179	0.73%
Monoclonal antibodies	0	0.00%
Other antibiotics	893	3.66%
Oxazolidinones	503	2.06%
Penicillin	482	1.97%
Penicillin with beta-lactamase inhibitors	6044	24.75%
Polymyxins	39	0.16%
Tetracyclines	811	3.32%
Combination therapy (Top 5 most frequent combinations)		
Cephalosporins and macrolides	5591	22.89%
Glycopeptides and penicillin with beta-lactamase inhibitors	4742	19.42%
Cephalosporins and glycopeptides	4360	17.85%
Cephalosporins, glycopeptides, and macrolides	1794	7.35%
Cephalosporins with other antibiotics	1596	6.54%

**Table 3 antibiotics-14-00642-t003:** Outcomes.

	Total N = 24,422				
	n	%	Mean	SD	Median
Index hospitalization outcomes					
Hospital mortality	3881	15.89%			
*Clostridioides difficile* infection	271	1.19%			
Total cost (USD)					
Hospital			USD 91,965	USD 133,734	USD 54,806
ICU			USD 23,847	USD 36,760	USD 12,063
LOS (days)					
Hospital			13.6	12.1	10.0
ICU			8.7	9.6	6.0
Daily cost (USD)					
Hospital			USD 6605	USD 18,371	USD 5412
ICU			USD 2684	USD 2247	USD 2120
Readmission outcomes					
30-day readmission					
Incidence	4082	19.87%			
CABP admission diagnosis	57	1.40%			
Cost, first readmission (USD)			USD 50,667	USD 83,279	USD 26,112
Length of stay, first readmission (days)			10.03	10.88	7
Cost, all readmissions (USD)			USD 61,072	USD 102,604	USD 30,851
LOS, all readmissions (days)			11.49	12.15	8
90-day readmission					
Incidence	6148	29.93%			
CABP admission diagnosis	67	1.09%			
Cost, first readmission (USD)			USD 48,927	USD 77,243	USD 25,932
Length of stay, first readmission (days)			9.49	10.16	7
Cost, all readmissions (USD)			USD 72,117	USD 113,421	$35,447
LOS, all readmissions (days)			13.25	14.16	9

CABP, community-acquired bacterial pneumonia; ICU, intensive care unit; LOS, length of stay.

**Table 4 antibiotics-14-00642-t004:** Risk factors for 30- and 90-day readmissions.

	30-Day Readmissions			90-Day Readmissions		
	Odds Ratio	95% Confidence Interval		Odds Ratio	95% Confidence Interval	
		Lower Limit	Upper Limit		Lower Limit	Upper Limit
Age (reference: 18–34 years)						
35–44 years	1.161	0.917	1.470	1.224	1.000	1.498
45–54 years	1.430	1.161	1.760	1.472	1.231	1.760
55–64 years	1.309	1.079	1.588	1.391	1.180	1.641
≥65 years	1.362	1.132	1.637	1.431	1.223	1.675
Female gender	1.041	0.971	1.116	1.028	0.967	1.093
Region (reference: Northeast)						
Midwest	1.073	0.918	1.254	1.007	0.875	1.159
South	0.825	0.739	0.922	0.840	0.762	0.926
West	0.823	0.731	0.927	0.865	0.779	0.960
Admission year (reference: 2018)						
2019	0.981	0.891	1.081	0.958	0.880	1.042
2020	0.980	0.891	1.078	0.955	0.878	1.037
2021	0.984	0.883	1.095	0.928	0.845	1.020
Diabetes mellitus	1.447	1.315	1.593	1.533	1.407	1.671
Chronic heart failure	1.441	1.288	1.613	1.517	1.369	1.680
Chronic obstructive pulmonary disease	1.427	1.288	1.581	1.471	1.340	1.613
ICU length of stay (per 1 day)	1.013	1.009	1.016	1.009	1.006	1.012

ICU, intensive care unit.

## Data Availability

The dataset analyzed in the current study was used under the auspices of IQVIA, and is available from the corresponding author on reasonable request and with permission from IQVIA.

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
