# Peer review of "The Burden of Hospitalization and Rehospitalization Among Patients Hospitalized with Severe Community-Acquired Bacterial Pneumonia in the United States, 2018–2022"

_antibiotics, 2025, doi:10.3390/antibiotics14070642_

Round 1
Reviewer 1 Report
Comments and Suggestions for Authors
Dear authors,
The article you have submitted is of interest to health professionals in general, and in particular to those who deal with the treatment of serious community-acquired respiratory infections.
In my opinion, I consider that the article is suitable for publication in another journal and not in the journal Antibiotics.
Therefore, I recommend that the authors resubmit it to another specialized journal (e.g. - American Journal of Respiratory and Critical Care Medicine or European Respiratory Journal).
Author Response
Comment: The article you have submitted is of interest to health professionals in general, and in particular to those who deal with the treatment of serious community-acquired respiratory infections.
In my opinion, I consider that the article is suitable for publication in another journal and not in the journal Antibiotics.
Therefore, I recommend that the authors resubmit it to another specialized journal (e.g. - American Journal of Respiratory and Critical Care Medicine or European Respiratory Journal).
AU: We appreciate the reviewer’s comment and suggestion. It is our feeling at this time that the study should be of interest to the Antibiotics readership. Should the Editors tell us otherwise, we will seek an alternate journal with this suggestion in mind.
Reviewer 2 Report
Comments and Suggestions for Authors
The authors present important data regarding severe bacterial pneumoniae requiring hospitalization. The study is overall well-conducted, the statistical analysis is appropriate, the results are well-structured and figures/tables are self-explanatory. There are no major grammatic or syntax errors. Below there are some comments that could possibly improve the manuscript:
- Please define Charlson Comorbidity Index score in Methods section.
- Statistical analyses: Please state the statistical program that was applied for analysis, the assumption of normality evaluation tests as well as when mean or median values were used, respectively.
- Please replace "Payer" with "Paying method" in Table 1.
- Table 2: Please replace "antibiotic class" with "antimicrobial class". Please define "beta-lactams" other than cephalosporins, since cephalosporins belong to beta-lactams group.
- Table 4: Please add p-values, if possible.
- Are the authors aware of other similar studies with cost estimates involved in Europe?
- It would be interesting to evaluate antimicrobial administration and cost differences before and during COVID-19 pandemic (2018-2019 and 2020-2021, respectively). Does COVID-19 pandemic play a significant factor in costs and mortality rates in regression analysis?
Author Response
The authors present important data regarding severe bacterial pneumoniae requiring hospitalization. The study is overall well-conducted, the statistical analysis is appropriate, the results are well-structured and figures/tables are self-explanatory. There are no major grammatic or syntax errors. Below there are some comments that could possibly improve the manuscript:
Comment 1: Please define Charlson Comorbidity Index score in Methods section.
AU: We have now added a citation (#19) to explain CCI.
Comment 2: Statistical analyses: Please state the statistical program that was applied for analysis, the assumption of normality evaluation tests as well as when mean or median values were used, respectively.
AU: The statistical program used was SAS v9.4, Cary, North Carolina. We have now added this to the Methods section. As for reporting mean vs. median, we chose the mean value for all continuous parameters so as to provide a way for modelers and policy makers to derive aggregate resource use associated with annual sCABP hospitalizations.
Comment 3: Please replace "Payer" with "Paying method" in Table 1.
AU: We are curious why the reviewer is requesting this change, as “Payer” is a standard rubric for who is responsible for paying the bill.
Comment 4: Table 2: Please replace "antibiotic class" with "antimicrobial class". Please define "beta-lactams" other than cephalosporins, since cephalosporins belong to beta-lactams group.
AU: We have changed to “antimicrobial.” As for beta-lactams, we have now removed that category altogether so as to avoid further confusion.
Comment 5: Table 4: Please add p-values, if possible.
AU: Most journals prefer 95% CIs to p values, and this is what we chose to provide. Additionally, a p value can be assumed not to have reached the alfa of 0.05 if the 95% CI crosses the value of 1.
Comment 6: Are the authors aware of other similar studies with cost estimates involved in Europe?
AU: We are aware of a handful of studies focusing on sCABP in the EU, of which only one (by Campling from 2022) engages with ICU costs, and none reports hospital costs.
Comment 7: It would be interesting to evaluate antimicrobial administration and cost differences before and during COVID-19 pandemic (2018-2019 and 2020-2021, respectively). Does COVID-19 pandemic play a significant factor in costs and mortality rates in regression analysis?
AU: We thank the reviewer for this suggestion, and are considering ways to answer this question outside the scope of the current study.
Reviewer 3 Report
Comments and Suggestions for Authors
This is a very interesting study of retrospective adults group cohort study on 5 years, identified CABP (CAP) remains one of the most costly causes of hospitalization) via an ICD-10 code algorithm. sCABP is an episode requiring ICU care, being developed models of identifying readmissions risk factors. 20% of cases required a readmission and were very costing. The strongest predictors of readmission were: older age and comorbidities. sCABP means a large population with high mortality and 30-day readmissions. It is a very big study - data sourced from 450 hospitals. The descriptive statistic was very good and developed regression models to predict risk of a 30- and 90 day readmission, with usefull tabless and graphs; and estimates the costs of annual readmissions. also showed a substantial risk of CDI development among patients hospitalized with CAP.
Author Response
Comment: This is a very interesting study of retrospective adults group cohort study on 5 years, identified CABP (CAP) remains one of the most costly causes of hospitalization) via an ICD-10 code algorithm. sCABP is an episode requiring ICU care, being developed models of identifying readmissions risk factors. 20% of cases required a readmission and were very costing. The strongest predictors of readmission were: older age and comorbidities. sCABP means a large population with high mortality and 30-day readmissions. It is a very big study - data sourced from 450 hospitals. The descriptive statistic was very good and developed regression models to predict risk of a 30- and 90 day readmission, with usefull tabless and graphs; and estimates the costs of annual readmissions. also showed a substantial risk of CDI development among patients hospitalized with CAP.
AU: We appreciate the reviewer’s attention to our work.
Round 2
Reviewer 2 Report
Comments and Suggestions for Authors
The authors only partially covered the reviewer's suggestions, therefore they should modify the text according to the initial comments. Below there are some further actions recommended for the improvement of the study in order to qualify for publication:
-
Clarify case definitions, ICD-10 codes, and handling of viral/COVID pneumonia.
-
No mention of model fit metrics (e.g., c-statistic, AUC). No sensitivity analysis to assess robustness (e.g., stratifying by year, hospital type). Potential misclassification and lack of time-to-event analysis (Cox model might better capture rehospitalization dynamics). No multilevel model to account for hospital-level clustering. Please report model diagnostics (e.g., AUC) and consider sensitivity or stratified analyses.
-
Streamline the discussion and avoid redundant explanations.
-
Ensure:
-
Numerical formatting: e.g., use “$91,965 (SD $133,734)” consistently.
-
Punctuation spacing: e.g., space before/after “%”, “,”, and parentheses.
-
Abbreviations: define all at first mention in both abstract and main text (e.g., CCI, sCABP).
-
Use of italics for p-values, e.g., p < 0.05.
-
Correct formatting/grammar issues and ensure consistency in tables/figures. More specifically:
-
Awkward phrasings:
-
“...who are severely ill” → consider “...presenting with severe illness”
-
“...add considerable costs to the overall care” → consider “...significantly increase total care costs”
-
-
Avoid passive constructions:
-
“The database is comprised of...” → use “The database comprises...”
-
-
Some repetition (e.g., on CDI and comorbidities) could be eliminated.
-
Streamline the Discussion to avoid redundancy across cited studies.
-
Table 1: Consider adding a “Total” row or aligning N=24,422 at the top. Please replace "Payer" with "Paying method"
-
Figure 1: Add axis labels (e.g., "Proportion of patients", "Number of readmissions").
-
Table 4: Consider color-coding or bolding significant ORs for clarity.
-
Use consistent citation style (e.g., Vancouver or IEEE).
-
Ensure all references are numbered consecutively and match in-text citations.
-
-
-
-
-
Consider a brief subgroup analysis (e.g., by year or age strata) to deepen insights.
-
Readmission data are well contextualized, but should: Distinguish planned vs. unplanned readmissions. Clarify proportion due to pneumonia recurrence vs. other causes. Explore cause-specific mortality/readmission where feasible.
Author Response
The authors only partially covered the reviewer's suggestions, therefore they should modify the text according to the initial comments. Below there are some further actions recommended for the improvement of the study in order to qualify for publication:
- Clarify case definitions, ICD-10 codes, and handling of viral/COVID pneumonia.
AU: As we state on page 2 in section 2.2 of the Methods,
“The case identification approach relied on International Classification of Diseases-10 (ICD-10) codes, excluding diagnosis of non-bacterial pneumonia during index hospitalization (see Supplemental Materials, File #1). Severe CABP was defined by the need for an intensive care unit (ICU) stay during the index hospitalization. To ensure that the index CABP hospitalization was not a repeat episode, we excluded all patients with any pneumonia within 90 days prior to index admission. Additional exclusion criteria were transfer from another hospital, missing or invalid year of birth, missing gender, or missing total charge or no revenue charges.”
As for the specific codes, those are listed in Supplemental File #1.
- No mention of model fit metrics (e.g., c-statistic, AUC). No sensitivity analysis to assess robustness (e.g., stratifying by year, hospital type). Potential misclassification and lack of time-to-event analysis (Cox model might better capture rehospitalization dynamics). No multilevel model to account for hospital-level clustering. Please report model diagnostics (e.g., AUC) and consider sensitivity or stratified analyses.
AU: We apologize, but we used the word “predictors” too loosely. The model we developed is an explanatory one, not a predictive one. We have now revised this point throughout the manuscript. Given that this is not a predictive model, the effort in our view did not merit model fit statistics. In fact, cross validation in a future study would be a better tool to establish our model’s generalizability.
We additionally, point the reviewer to other published papers where modeling does not report fit statistics. Here is an example of such a recent paper published in this Journal:
Papiol E, Berrueta J, Ruíz-Rodríguez JC, Ferrer R, Manrique S, Claverias L, García-Martínez A, Orts P, Díaz E, Zaragoza R, et al. Does Empirical Antibiotic Use Improve Outcomes in Ventilated Patients with Pandemic Viral Infection? A Multicentre Retrospective Study. Antibiotics. 2025; 14(6):594. https://doi.org/10.3390/antibiotics14060594
- Streamline the discussion and avoid redundant explanations.
- Ensure:
- Numerical formatting: e.g., use “$91,965 (SD $133,734)” consistently.
- Punctuation spacing: e.g., space before/after “%”, “,”, and parentheses.
- Ensure:
AU: We would appreciate if the reviewer could point us to specific places where we have introduced redundancies and/or formatting errors.
- Abbreviations: define all at first mention in both abstract and main text (e.g., CCI, sCABP).
AU: We do not use the abbreviation CCI in the text, and, as far as we can see, we have defined all appropriate abbreviations at their first mention. If we have made any errors, we would appreciate specific places where we have done so. Thank you.
- Use of italics for p-values, e.g., p< 0.05.
AAU: We will defer to the Journal’s preferred formatting.
- Correct formatting/grammar issues and ensure consistency in tables/figures. More specifically:
- Awkward phrasings:
- “...who are severely ill” → consider “...presenting with severe illness”
- Awkward phrasings:
AU: We appreciate the reviewer’s stylistic suggestions and will await further instruction from the Editors as to their preferences.
- “...add considerable costs to the overall care” → consider “...significantly increase total care costs”
AU: Same as above
- Avoid passive constructions:
- “The database is comprised of...” → use “The database comprises...”
AU: Same as above
- Some repetition (e.g., on CDI and comorbidities) could be eliminated.
AU: We would appreciate further guidance from the reviewer regarding where these repetitions are.
- Streamline the Discussion to avoid redundancy across cited studies.
AU: Again, we would appreciate specific guidance on this. Thank you.
- Table 1: Consider adding a “Total” row or aligning N=24,422 at the top.
AU: We have not relabeled the column Total N = 24,422 in each table.
- Please replace "Payer" with "Paying method"
AU: As we said in our previous responses, we are not sure why this is necessary, as “payer” or “payor” is a routine way to refer to who is the payer.
- Figure 1: Add axis labels (e.g., "Proportion of patients", "Number of readmissions").
AU: In our copy of the paper, the axis labels for Figure 1 are “Proportion of patients” for the y-axis, and Readmissions for the x-axis. The legend spells out “Number of readmissions.”
- Table 4: Consider color-coding or bolding significant ORs for clarity.
AU: While we did bold significant ORs, we will defer to the editorial preferences if/when the paper is accepted. Thank you.
- Use consistent citation style (e.g., Vancouver or IEEE).
AU: We will defer to the editorial preferences.
- Ensure all references are numbered consecutively and match in-text citations.
AU: As far as we can tell, this is the case.
- Consider a brief subgroup analysis (e.g., by year or age strata) to deepen insights.
AU: We agree with the reviewer that there are a number of other analyses that would lend deeper insights into this population. However, the aim of the current study was to characterize overall population, and that is what is presented in the manuscript.
- Readmission data are well contextualized, but should:
- Distinguish planned vs. unplanned readmissions.
AU: Unfortunately, this is not feasible in administrative data
- Clarify proportion due to pneumonia recurrence vs. other causes.
AU: These numbers can be found in Table 3 and in Supplemental File #6.
- Explore cause-specific mortality/readmission where feasible.
AU: This is also not feasible.
Round 3
Reviewer 2 Report
Comments and Suggestions for Authors
The comments of the authors are fine. The manuscript fulfills all the requirements for publication.